# Presence, Location and Conservation of Putative G-Quadruplex Forming Sequences in Arboviruses Infecting Humans

**DOI:** 10.3390/ijms24119523

**Published:** 2023-05-30

**Authors:** Giulia Nicoletto, Sara N. Richter, Ilaria Frasson

**Affiliations:** Department of Molecular Medicine, University of Padua, Via A. Gabelli 63, 35121 Padua, Italyilaria.frasson@unipd.it (I.F.)

**Keywords:** arthropod-borne viruses, G-quadruplex, innovative targeting, prediction of non-canonical RNA structures

## Abstract

Guanine quadruplexes (G4s) are non-canonical nucleic acid structures formed by guanine (G)-rich tracts that assemble into a core of stacked planar tetrads. G4s are found in the human genome and in the genomes of human pathogens, where they are involved in the regulation of gene expression and genome replication. G4s have been proposed as novel pharmacological targets in humans and their exploitation for antiviral therapy is an emerging research topic. Here, we report on the presence, conservation and localization of putative G4-forming sequences (PQSs) in human arboviruses. The prediction of PQSs was performed on more than twelve thousand viral genomes, belonging to forty different arboviruses that infect humans, and revealed that the abundance of PQSs in arboviruses is not related to the genomic GC content, but depends on the type of nucleic acid that constitutes the viral genome. Positive-strand ssRNA arboviruses, especially Flaviviruses, are significantly enriched in highly conserved PQSs, located in coding sequences (CDSs) or untranslated regions (UTRs). In contrast, negative-strand ssRNA and dsRNA arboviruses contain few conserved PQSs. Our analyses also revealed the presence of bulged PQSs, accounting for 17–26% of the total predicted PQSs. The data presented highlight the presence of highly conserved PQS in human arboviruses and present non-canonical nucleic acid-structures as promising therapeutic targets in arbovirus infections.

## 1. Introduction

Vector-borne diseases are bacterial, viral, or parasitic infections transmitted to the human host through the bite of infected arthropod species, such as mosquitoes, ticks, midges, and flies [1]. In 2020, the World Health Organization (WHO) declared that vector-borne diseases accounted for approximately 20% of all infectious diseases [2]. In the case of viral infections, arthropod-borne virus (arbovirus) infections are a global threat, as travel and trade contribute to the spread of vectors and viruses over large geographical areas, and climate change favours disease transmission [3,4,5]. Arboviruses are a large group of RNA viruses belonging to different families and genera, of which about fifty members are known to infect humans. A few members of arboviruses cause mild flu-like symptoms and joint pain. The vast majority of Arboviruses cause severe and life-threatening disease, with mortality rates as high as 50% [2,6,7]. Specific anti-arbovirus treatments are not available and vaccines have been developed against less than 10% of the arboviruses. As a result, arbovirus infections are controlled solely by prevention strategies to hinder the spread of viruses in the environment and among humans. Efforts to prevent and treat vector-borne viral diseases must be intensified. In fact, in 2022, WHO launched the Global Arbovirus Initiative to promote all initiatives to control arboviruses with epidemic and pandemic potential [8].

Approved antiviral drugs target viral proteins involved in key viral steps, from viral entry to viral gene expression and genome replication. Direct targeting of nucleic acid is very sporadic, as achieving selective targeting has always been extremely challenging. Nucleic acids have been shown to fold into structures alternative to the classical double helix, which do not obey the Watson-Crick hybridization canon and are therefore defined as non-canonical nucleic acid structures. Among these non-canonical structures, G-quadruplexes (G4s) have been shown to play key biological roles both at the human and viral level [9,10,11,12,13,14,15]. G4s can form in G-rich sequences of DNA or RNA, in which four guanines (Gs) are linked by Hoogsteen-type hydrogen bonds to form planar square structures called G-quartets. The stacking of successive G-quartets leads to the formation of the G4 structure, which is supported and stabilized by physiological cations, such as potassium or sodium [16]. G4s have been identified primarily in mammalian cells, but more recently their presence in viruses, bacteria and parasites has also been investigated [17]. At the viral level, G4s are involved in the control of key viral processes, such as transcription, genome replication and the induction or maintenance of the viral latency [12].

Several algorithms have been validated to predict the presence and distribution of putative quadruplex (G4)-forming sequences (PQSs) in genomes [18]. The different algorithms calculate the presence of PQSs or the G4 folding propensity, taking into account the number of Gs and G islands as well as the loop length. G4 prediction algorithms have been trained on the human genome [18,19]. To date, few bioinformatic analyses have predicted PQSs in microorganisms using tools such as the well-established QGRS and G4Hunter [18,19,20,21,22,23,24,25]. Viral genomes have been shown to contain G4s that do not strictly follow the rules of canonical G4s, but include bulges, mismatches and stem loops [26,27]. Therefore, PQS prediction algorithms that take into account the possibility of G4s folding from imperfect G-runs should be used to better estimate the presence of PQSs on viral genomes. Recently, Bioconductor’s pqsfinder tool was released as a flexible tool for analyzing putative PQSs that also contain bulges or mismatches [28]. 

This work shows that arboviruses embed both canonical and bulged PQSs and that the different viral families show different patterns of PQSs enrichment or depletion. The conservation of each predicted PQS among virus isolates was also analysed to correlate the presence of highly conserved putative G4 sequences with their possible biological role. Our data provide new information on the evolutionary conserved PQSs among human arboviruses, provide insights into unexplored aspects of arbovirus biology and reveal innovative anti-arbovirus targets.

## 2. Results

### 2.1. Prediction of PQSs in Human Arboviruses

Arboviruses were grouped according to the Expasy ViralZone and NCBI taxonomy classifications [29,30]. A total of 40 different arboviruses infecting humans was retrieved, which were further divided into three groups on the basis of the type of nucleic acid constituting their genomes: 1 dsRNA, 16 negative-strand ssRNA and 23 positive-strand ssRNA (Table 1). For each virus, the complete set of sequenced genomes was downloaded from the NCBI database. Partially sequenced and unverified genomes, as well as genomes containing nucleotide strings longer than five nucleotides without base assignment (i.e., NNNNN) were not considered for further analysis. For each virus, the nucleotide sequence to be considered as reference genome was retrieved from the NCBI Reference repository. The accession numbers are listed in Table 1. 

First, for each virus, the GC content of the reference genome alone and of all the sequenced virus isolates was calculated and expressed as an average value (Table 1). Taken together the arboviruses have an average GC content of 44%. Positive-strand ssRNA viruses reference genomes have an average GC content around 50% (48–55%), whereas negative-strand ssRNA viruses reference genomes display GC contents that span from 33% to 50%. The Banna virus, which is the only arbovirus with a dsRNA segmented genome, has a GC content of 37% to 42%, depending on the segment. Analysis of the mean GC content of all sequenced genomes per virus provided data on the conservation of G and C residues, possibly involved in G4 formation. For the majority of positive-strand ssRNA viruses, the GC content was conserved, with the exception of Dengue strains 1 and 2. Notably, Dengue 1 and 2 are also the viruses with the highest number of sequenced genomes among all analyzed viruses, 2095 and 1764, respectively. The analysis of the negative-strand ssRNA viruses showed that the family members with segmented genomes shared a lower GC content conservation (e.g., Crimean Congo hemorrhagic fever virus), whereas the members with the genome composed of a single linear molecule of RNA (e.g., Chandipura virus) showed a very high GC content conservation. Among segmented negative-strand ssRNA viruses, the S segment, coding for non-structural proteins, was the less conserved. Once again, the viruses with the highest numbers of sequenced genomes showed the highest variability in GC content values. The conservation of the GC content in the segments of the Banna virus, the only dsRNA virus in this analysis, was very segment-dependent. Segment 9 was the less conserved and codes for the outer-capsid protein VP9. In this case too, we could analyze more sequences from this segment than from the other eleven. This may be because VP9 has been studied and recognized as the protein involved in host attachment and viral internalization [31].

Next, the pqsfinder algorithm was run on all reference genomes. The algorithm was set up to recognize sequences with G-runs containing at least two G residues. Each PQSs could have loops with a maximum length of 12 nucleotides and a maximum of one loop with a length of zero nucleotides. The pqsfinder algorithm was used to identify canonical PQSs and PQSs harboring a single bulge (Table 2). PQSs containing mismatches (non-G bases in the G-quartet) were excluded from the prediction. The minimum acceptable score was set at 12, in order to exclude PQSs that were characterized by short G-runs, together with long loops and a bulge, and therefore unlikely to form. Both the positive and the negative RNA strands were analyzed for the presence of PQS, as they represent two different stages of viral infection and are both essential in the viral replication cycle (Table 2) [32]. 

The analysis that considered canonical and bulged PQSs showed that the reference genomes of positive-strand ssRNA viruses, Flavivirus and Alphavirus, are particularly enriched in PQSs. In particular, the Japanese encephalitis, the Langat, the Louping ill, the Tick-borne encephalitis and Zika viruses were predicted to embed more than one hundred PQSs in their genomes. In general, the members of the Flavivirus family embed an average of 94 PQSs per genome. The second group of positive-strand RNA viruses, the Alphaviruses have an average of 67 PQSs per genome, with the Semliki Forest virus topping the list with 95 PQSs. Negative-strand ssRNA viruses, with the genome consisting of a single linear RNA strand, showed an average of 37 PQSs, with the Chandipura virus and the Australian bat lyssavirus showing the highest number of PQSs (i.e., 46 and 45, respectively) and the non-Indiana Vesicular stomatitis virus strains showing the fewest (i.e., 29). Segmented negative-strand ssRNA viruses and the dsRNA Banna virus, although not so different in GC content from the other viruses, were predicted to have very few PQSs (average of 10 and 2, respectively). A closer look at the PQSs strand location (Appendix A) showed that among the positive-strand RNA viruses, Flaviviruses embed more PQSs in the positive strand (i.e., the viral genome, but also the viral mRNA) [33], whereas Alphaviruses have members with equal strand distribution (Chikungunya, Mayaro, O’nyong-nyong, Ross River, Sagiyama, Semliki Forest, Venezuelan and Western equine encephalitis) and members with PQSs mainly located in the antigenome strand (negative-strand) [34]. Single linear negative-strand RNA viruses have more PQSs in the positive strand, i.e., in the antigenome which corresponds also to the viral mRNA. Segmented negative-strand viruses showed the highest variability, with PQS distributed on both strands, depending on the virus.

The number of canonical PQSs was then calculated, excluding the bulged sequences from the first prediction (Table 2). The maximum loop length and minimum sequence score remained the same as in the previous analysis. This analysis showed that approximately 22% of the calculated PQSs are non-canonical, when arboviruses are considered as a single group. Looking at single classes of viruses, 21% of PQSs in Alphaviruses (positive-strand ssRNA viruses), 26% in negative-strand ssRNA viruses with a single linear RNA and 23% in segmented RNA viruses contain a bulge. Flaviviruses, the viruses with the highest number of PQSs, are less likely to have bulged PQSs (17%). The Banna virus has 22% of non-canonical PQSs (Table 2).

Next, the significance of the predicted PQSs was then calculated. To assess whether the predicted PQSs in arboviruses were statistically relevant or random, the results from viral genomes were compared with those obtained by viral genome simulation. Shuffled genomes (one hundred per virus), with the same nucleotide composition but different order with respect to the references, were generated. The presence of PQSs was predicted using the same parameters as in the first analyses (Table 2). To estimate the statistical significance of PQSs prediction, data on viruses (Reference genomes) and on shuffled genomes were subjected to one-sample *t*-test and *p*-values were calculated [35]. The one-sample *t*-test was used to determine whether the average PQSs number of the shuffled genomes (one hundred per virus) was significantly different from the PQSs number predicted on the relative viral reference genome. *p*-values lower than 0.001 were considered significant. Significance analysis and generation of corresponding *p*-values indicated that 52% of the considered viral genomes/segments were significantly enriched in PQSs, while 37% showed significant depletion in PQSs compared to the presence of G-runs on shuffled genomes/segments. The remaining 11% of viral genomes/segments showed no significant enrichment or depletion in PQSs. The PQS prediction was highly significant for positive-strand RNA viruses, with the exception of the Eastern equine encephalitis virus and the Ross River virus. Flaviviruses are all enriched in PQSs, whereas Alphaviruses, with the exception of Barmah Forest and Semliki Forest viruses, are depleted in PQSs. When considering negative-strand RNA genomes, viruses with single linear RNA genomes are all statistically enriched in PQSs, whereas segmented viruses display segments with PQSs enrichment, PQSs depletion and segments that are not significantly enriched in PQSs (i.e., Rift Valley fever virus). The Banna virus (dsRNA) has segments 10 and 11 with non-significant *p*-values, the other 10 segments have a statistically significant PQS prediction, despite the low number of predicted PQSs.

### 2.2. Conservation of Predicted PQSs and Genomic Location of Highly Conserved PQSs

Once the presence of PQSs in arboviruses had been predicted and their statistical relevance had been assessed, their distribution within the genomes was examined (Figure 1 and Appendix A, density panels). PQS density distribution was calculated using pqsfinfer density function: it indicates if and where PQSs were clustered within the genome of interest; high scoring PQSs clustered in high density regions are considered to have a higher folding potential. We observed that, in general, PQSs were widely distributed across the length of the genome and that PQSs with high scores, and therefore more likely to form, tended to cluster together.

RNA viruses are prone to genomic mutations to enhance their environment/host adaptability [36,37], so the conservation of PQSs across all sequenced isolates of each virus species was assessed, hypothesizing that the presence of a conserved PQS in a poorly conserved genomic environment would strengthen the hypothesis of a significant biological function. The conservation rate of each predicted PQSs in all genome/segment sequences we retrieved from the NCBI database was calculated (Figure 1 and Appendix A). We considered the conservation analysis to be significant when at least 5 isolates per virus were available.

The different RNA virus populations have different mutation rates [38]. Positive-strand ssRNA viruses have high mutation rates, followed by negative-strand ssRNA viruses. DsRNA viruses have the lowest mutation rate of the three viral classes analyzed. Notably, PQSs do not seem to follow this rule, as many members of the Flavivirus and Alphavirus (e.g., West Nile virus and Semliki Forest virus) have highly conserved PQSs throughout the genome. The Eastern equine encephalitis and the Ross River viruses, which would be excluded by the previously calculated *p*-value on shuffled genomes, have highly conserved PQSs. On the contrary, few negative-strand ssRNA viruses have highly conserved PQSs. We found few conserved PQSs in the segments of the Uukuniemy and the Sandfly viruses and in the genomes of the Chandipura and Vesicular stomatitis viruses. In the case of segmented negative-stranded RNA viruses, we found that segments with significant *p*-values (Table 2) did not have non-significantly conserved PQSs. The Banna virus (dsRNA) was not only poor in terms of predicted PQSs, although they were statistically significant, but also showed a very low rate of PQSs conservation among the virus isolates. Taken together, the conservation analysis revealed that the conservation of a particular PQS among isolates belonging to the same virus was not related to the initial prediction score (i.e., the folding propensity and associated stability), nor did it depend on the presence of other PQSs in the vicinity (density).

The conservation of canonical and bulged PQSs was calculated and their percentage of conservation was reported if they were conserved in more than 80% of the analysed viral genomes. Notably, in positive polarity viruses (Flaviviruses and Alphaviruses) bulged and canonical PQSs were conserved to the same extent, with viruses such as the Sagiyama and the Semliki Forest viruses having more than 87% of fully conserved canonical and bulged PQSs. The pattern of conservation among negative polarity RNA viruses is much more diverse. Segmented RNA viruses do not appear to conserve bulged PQSs, nor do they have a high conservation rate of canonical PQSs. Single linear negative polarity RNA viruses tend to conserve both canonic and bulged PQSs, with viruses such as the Australian bat Lyssavirus with no conservation potential and viruses such as the Isfahan virus and the Vesicular stomatitis (non-Indiana strains) having high conservation rates of both canonical and bulged PQSs. The Banna virus (dsRNA) showed no strong conservation of either type of PQSs (Table 2). 

The genomic location of all PQSs with more than 85% conservation was then determined (Figure 1 and Appendix A, conservation panel, Figure 2 and Appendix A). Genome coordinates were obtained for 5′- and 3′-untranslated (UTR) and coding sequences (CDS). For the majority of arboviruses, the CDSs are well defined in the Reference genomes, whereas the UTRs are more inconsistently annotated. When missing in the annotation file, the UTRs were manually defined as the regions preceding the first CDS and closing the genome after the last nucleotide of the last CDS. Flaviviruses showed a strong tendency to have and preserve PQSs in coding regions but also in 3’ UTRs. The vast majority of conserved PQSs of Alphaviruses are embedded in coding sequences, with the exception of the Semliki Forest and Ross River viruses, which also have conserved PQSs in the UTRs. Negative-strand viruses preserve PQSs in coding sequences, with exceptions such as Bunyamwere La Crosse virus, Dugbe virus, Sadfly Sicilian virus and Rift Valley fever virus that embed conserved PQSs also in UTRs of L and M segments (Figure 2 and Appendix A). 

## 3. Discussion

The presence and possible key roles played by nucleic acid secondary structures, in particular G4s, during the viral cycle of major human pathogens, such as HIV-1, HSV-1 and many others [12], has begun to be demonstrated. Understanding the regulation of G4 folding in viruses has attracted much attention due to the potential use of G4s as targets for innovative antiviral therapies [14]. The most comprehensive viral genome analyses have been performed using algorithms that predict canonical putative G4s [18,22,39], or using algorithms that penalize sequences with cytosine runs [20,21,23]. These pattern-based algorithms do not consider non-canonical G4 forming sequences, such as those containing bulges or stem loops. Notably, both G4s folding with bulges or forming stem loops have been reported at the viral level [26,27,40]. 

Arboviruses are a major threat to humans with no specific pharmacological treatment and few prevention strategies [4]. Here we challenged the PQS Finder algorithm, which was designed to be imperfection-tolerant and validated on the human sequence data [28], with the genomes of arbovirus that infect humans. We had previously performed an extensive analysis on human viral pathogens using a traditional approach [39]. In this study we extended and included seven novel members of the arboviruses (Bunyavirus snowshoe hare, Chandipura, Dhori, Isfahan, Punta Toro, Sandfly fever Sicilian, Tick-borne encephalitis, Possawan encephalitis, and Usutu viruses) and examined the presence of PQSs and their conservation in all sequenced) virus isolates (up to February 2023). More than twelve thousand genomes/segments were analysed, belonging to the forty different arbovirus that infect humans. The present work provides new data showing that: a. arboviruses harbor bulged PQSs in addition to canonical ones; b. the vast majority of the predicted PQSs are statistically significant when compared with shuffled sequences; c. not only the canonical but also the bulged PQSs are conserved; d. the conserved PQSs are mainly located in coding sequences and 3’UTRs. 

Our data show that frequency of PQSs is not related to the GC content of viral genomes, confirm that the clustering of G-runs is not random, and suggest a specific biological role for G4 structures at the arboviral level. Viruses, especially those with an RNA genome, such as that of arboviruses, mutate with high frequency [36,38,41]. The comparison with randomly generated shuffled genomes showed that members of the arbovirus family are statistically enriched or depleted in PQSs, revealing that certain members of this RNA virus family, prevent the generation of novel regions that could fold into G4s, despite their high mutation rate. 

The conservation of PQSs in viruses is one of the strongest indications of the biological relevance of G4s. The utmost conservation of G-tracts in a G4-forming pattern indicates that they are required for infection/replication/transmission of the virus. Our data show that, depending on the virus class, both canonical and bulged PQSs are conserved among virus isolates, suggesting that also bulged G4s play a role in the biology of arboviruses. In addition, our analysis suggests that PQSs in the coding sequences and 3’UTRs of positive-strand ssRNA viruses, especially Flaviviruses, could play an essential role during viral infection.

The role of G4s in regulating transcription and translation when embedded in coding regions has begun to be elucidated at the human level [42]: our data emphaticize their regulatory role also in arboviruses, especially Flaviviruses where PQSs are mainly located on the positive RNA strand, which acts as both viral genome and viral mRNA. Furthermore, since the 3′ UTR of positive-strand ssRNA viruses regulates numerous aspects of the viral life cycle such as replication/translation and the complex network of the host-cell interactions, the highlighted presence of several G4s at this genomic level paves the way for a deeper understanding of G4s as regulators of novel aspects of arbovirus infection. 

Our data also show that G4s are not particularly abundant or conserved in negative-strand ssRNA or dsRNA arboviruses. Negative-strand and dsRNA viruses have more complex viral cycles than positive-polarity ssRNA viruses: under these conditions, G4-mediated slowing of viral transcription and replication may be more likely to be avoided [43].

## 4. Materials and Methods

### 4.1. Viral Genomes Selection

Accessible viral genomic sequences (12240 in total) belonging to the 40 arboviruses infecting humans were downloaded from the genome database of the NCBI in January 2023. Genomes were grouped in FASTA-format files. Dengue viruses were clustered according to the four reported serotypes (Dengue virus 1–4) [44]. Vesicular stomatitis viruses (VSVs) were divided into two separated groups, the first including the Indiana serotype and the second containing all the other serotypes [45]. For each virus, the NCBI entry code indicated in the NCBI reference repository was considered as reference genome [46]. For the West Nile virus, two different reference genomes (lineage 1 and Kunjiun subtype) are considered as reference, the NCBI Reference Sequence NC_009942.1 (lineage 1) was set as reference genome. Differently from Dengue virus isolates, West Nile sequences are not registered indicating the lineage or the subtype, so that all genomes corresponding to the West Nile virus were aligned together. FASTA files were purged from unverified or partially sequenced genomes, as well as from genome sequences containing multiple stretches of nucleotides lacking base assignments (i.e., NNN). NCBI accession codes and FASTA files containing all viral complete genomic sequences are shown in Table 1 and contained in Appendix A, respectively. Genomes were aligned using the Jalview platform [47].

### 4.2. Bioinformatic Prediction of Putative G4-Forming Sequences and Conservation Analysis

#### 4.2.1. Prediction of PQS

All the analyses were performed using R (version 4.2.2). The FASTA files containing the reference genomes and the FASTA files containing the multiple alignments were loaded onto the R platform and GC content was calculated using Biostrings (2.66) [48,49]. PQS prediction was performed on the reference genome using pqsfinder [28] (version 2.14.1) with the following parameters: deep = TRUE, min_score = 12, max_bulges = 1, max_mismatches = 0, loop_max_len = 12. The deep parameter has been set to TRUE to allow detection of PQS clusters. Canonical PQS prediction was performed retrieving the sequences displaying no bulges from the initial PQS prediction. PQS density was predicted using the pqsfinder density function. The PQS score was automatically assigned by the pqsfinder algorithm. All R scripts created to generate predictions, density, score, GC content, and conservation rates are available in the Appendix A.

#### 4.2.2. Shuffling and Statistical Analyses

The R universalmotif (version 1.16.0) was used to shuffle the reference genome sequences [50]. Each reference was shuffled one hundred times using the linear method with k-let = 1. For each shuffled sequence, a PQS prediction was performed with the pqsfinder parameters indicated in Section 4.2.1. To estimate the significance of the analysis, the one-sample *t*-test was performed comparing the PQS predictions of the shuffled genomes with the PQS prediction of the reference genome. Significance was expressed as a *p*-value.

#### 4.2.3. Conservation of PQS

To calculate the percentage of conservation of each PQS the vmatchPattern function of Biostrings (version 2.66) was used, setting the parameter with.indels = TRUE, to count PQS with longer loops as conserved. The multiple aligned genomes were loaded onto the R platform and the number of times each predicted PQS was present in the aligned genomes was counted. The percentage of conservation was also calculated, taking into account the number of aligned genomes analysed. The conservation values were then plotted together with the density pattern and the PQS scores using Gviz (version 1.42.0) [51].

#### 4.2.4. Annotation of Conserved PQS

Only PQSs with more than 80% conservation were annotated on viral genomes. GTF files were uploaded using rtracklayer (version 1.58.0) [52]. Annotation was performed using annotatr (version 1.24.0), with the length of each conserved PQS set as the minimum overlap [53]. The region preceding the first CDS was considered the “5’UTR”, while the region following the last CDS was considered the “3’UTR”. Bar graphs were generated using ggplot2 (version 3.4.0). 

## 5. Conclusions

Arboviruses are a heterogeneous family of viruses that are transmitted to humans by arthropod vectors. This work has shown that many members of the family, mainly belonging to the Flavivirus subgroup, embed highly conserved PQSs in their genomes. The conserved PQSs are located in coding regions and at genome ends, reinforcing the critical role of G4s in the regulation of viral cycles. These findings pave the way for a broader understanding of the mechanisms regulating arbovirus infections and suggest that highly conserved PQSs may be novel and innovative antiviral targets against arbovirus infections.

## Figures and Tables

**Figure 1 ijms-24-09523-f001:**
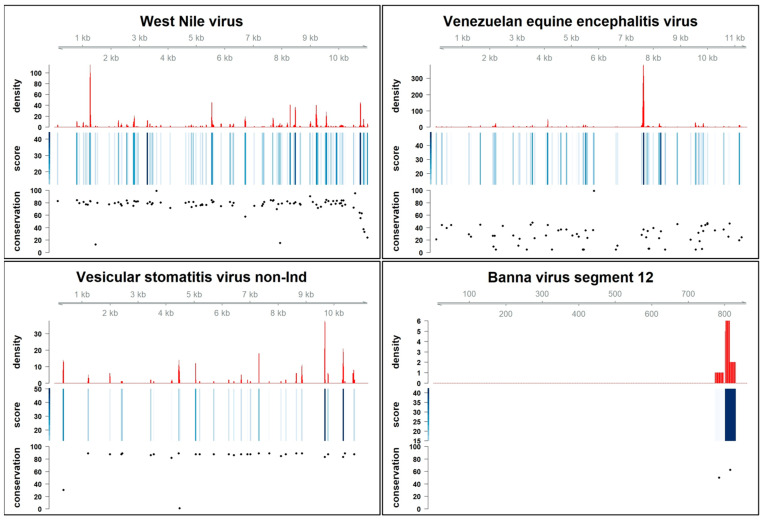
Presence, density, score and conservation of PQSs in arboviruses. Plots representing the PQS density (red bars), the score (blue bars) and the conservation percentage (black dots) of each predicted PQS. The viral genome length is reported above the density plot. The Vesicular stomatitis virus non-Indiana strains have been abbreviated to Vesicular stomatitis virus non-Ind.

**Figure 2 ijms-24-09523-f002:**
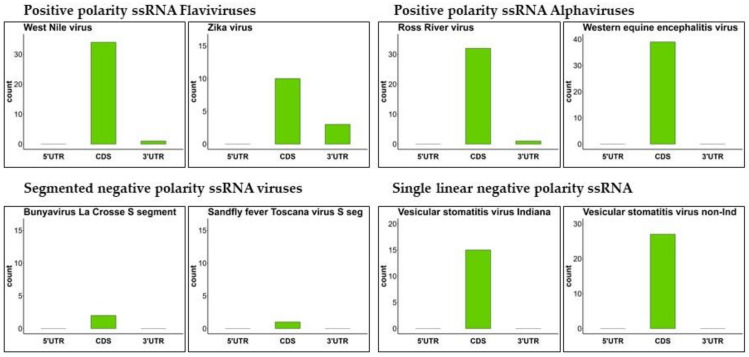
Conserved Genomic localization of PQSs in arboviruses. Plots reporting the annotation of highly conserved PQSs. Each viral genome was divided into three regions: Untranslated regions (5′ and 3′ UTRs) and coding sequences (CDS). PQSs were annotated on the basis of the official NCBI annotation of each viral Reference genome. In panels with long virus names, the word “segment” has been abbreviated to “seg”. The Vesicular stomatitis virus non-Indiana strains has been abbreviated to Vesicular stomatitis virus non-Ind.

**Table 1 ijms-24-09523-t001:** Analysed arboviruses data. The table reports the analysed viruses in alphabetical order. The columns indicate the virus name (Virus), the viral Genus/Family each virus belongs to (Genus, Family), the viral genomic nucleic acid type (Genome), the viral genomic structure (Genome structure), the name of each segment in case of segmented genomes (Segments), the analysed reference genome NCBI entry (Reference genome), the total number of analysed genomes/segments (Total analysed genomes and segments), the number of analysed sequences of each segment (Analysed segments), the average GC content of the Reference genomes and of the entire group of analysed genomes per virus (% GC Reference genomes and % GC all analysed genomes, respectively). Colors correspond to dsRNA (grey), negative-strand RNA ((-)ssRNA, yellow), positive-strand RNA ((+)ssRNA, blue) viruses. In segmented RNA viruses, the viral segments S, M and L are ordered by length.

Virus	Genus, Family	Genome	Genome Structure	Segments	Reference Genome	Total Analysed Genomes and Segments	Analysed Segments	% GC Reference Genomes	% GC All Analysed Genomes
Australian bat lyssavirus	Lyssavirus, Rhabdoviridae	(-)ssRNA	Single linear RNA		NC_003243.1	34		44	43
Banna virus	Seadornavirus, Reoviridae	dsRNA	12 Segmented RNAs	Segment 1	KC954611.1	128	7	38	39
				Segment 2	KC954612.1		7	40	40
				Segment 3	KC954613.1		9	40	37
				Segment 4	KC954614.1		8	40	39
				Segment 5	KC954615.1		7	40	39
				Segment 6	KC954616.1		10	42	40
				Segment 7	KC954617.1		12	37	35
				Segment 8	KC954618.1		8	43	42
				Segment 9	KC954619.1		37	38	32
				Segment 10	KC954621		8	38	37
				Segment 11	KC954621.1		7	39	39
				Segment 12	KC954622.1		8	38	38
Barmah Forest virus	Alphavirus, Togaviridae	(+)ssRNA	Single linear RNA		NC_001786.1	39		48	48
Bunyamwera virus	Orthobunyavirus, Bunyaviridae	(-)ssRNA	3 Segmented RNAs	Segment S	NC_001927.1	21	8	42	40
				Segment M	NC_001926.1		7	37	36
				Segment L	NC_001925.1		6	33	33
Bunyavirus La Crosse	Orthobunyavirus, Bunyaviridae	(-)ssRNA	3 Segmented RNAs	Segment S	NC_004111	100	39	41	40
				Segment M	NC_004109.1		34	38	38
				Segment L	NC_004108.1		27	35	35
Bunyavirus snowshoe hare	Orthobunyavirus, Bunyaviridae	(-)ssRNA	3 Segmented RNAs	Segment S	NC_055198.1	12	5	45	40
				Segment M	NC_055197.1		4	39	38
				Segment L	NC_055196.1		3	35	35
Chandipura virus	Vesiculovirus, Rhabdoviridae	(-)ssRNA	Single linear RNA		NC_020805.1	7		42	42
Chikungunya virus	Alphavirus, Togaviridae	(+)ssRNA	Single linear RNA		NC_004162.2	899		36	36
Crimean-Congo hemorrhagic fever virus	Nairovirus, Bunyaviridae	(-)ssRNA	3 Segmented RNAs	Segment S	NC_005302.1	642	211	46	40
				Segment M	NC_005302		196	43	34
				Segment L	NC_005301.3		235	41	38
Dengue virus 1	Flavivirus, Flaviviridae	(+)ssRNA	Single linear RNA		NC_001477.1	2095		47	44
Dengue virus 2		(+)ssRNA	Single linear RNA		NC_001474.2	1764		46	43
Dengue virus 3		(+)ssRNA	Single linear RNA		NC_001475.2	992		47	46
Dengue virus 4		(+)ssRNA	Single linear RNA		NC_002641	257		47	46
Dhori virus	Thogotovirus, Orthomyxoviridae	(-)ssRNA	6 Segmented RNAs	Segment 1	NC_034261.1	39	6	45	45
				Segment 2	NC_034263.1		7	45	45
				Segment 3	NC_034254.1		6	44	44
				Segment 4	NC_034255.1		7	48	47
				Segment 5	NC_034262.1		6	48	48
				Segment 6	NC_034256.1		7	49	49
Dugbe virus	Nairovirus, Bunyaviridae	(-)ssRNA	3 Segmented RNAs	Segment S	NC_004157.1	14	7	43	18
				Segment M	NC_004158.1		3	42	41
				Segment L	NC_004159.1		4	39	39
Eastern equine encephalitis virus	Alphavirus, Togaviridae	(+)ssRNA	Single linear RNA		NC_003899.1	455		49	49
Isfahan virus	Vesiculovirus, Rhabdoviridae	(-)ssRNA	Single linear RNA		NC_020806.1	2		42	42
Japanese encephalitis virus	Flavivirus, Flaviviridae	(+)ssRNA	Single linear RNA		NC_001437	328		51	51
Langat virus	Flavivirus, Flaviviridae	(+)ssRNA	Single linear RNA		NC_003690	3		54	54
Louping ill virus	Flavivirus, Flaviviridae	(+)ssRNA	Single linear RNA		NC_001809	28		55	55
Mayaro virus	Alphavirus, Togaviridae	(+)ssRNA	Single linear RNA		NC_003417.1	41		50	49
Murray Valley encephalitis virus	Flavivirus, Flaviviridae	(+)ssRNA	Single linear RNA		NC_000943	17		49	49
O’nyong-nyong virus	Alphavirus, Togaviridae	(+)ssRNA	Single linear RNA		NC_001512.1	7		48	48
Oropouche virus	Orthobunyavirus	(-)ssRNA	3 Segmented RNAs	Segment S	NC_005777.1	174	59	47	41
				Segment M	NC_005775.1		57	35	35
				Segment L	NC_005776.1		58	34	35
Punta Toro phlebovirus	Phlebovirus, Bunyaviridae	(-)ssRNA	3 Segmented RNAs	Segment S	DQ363406.1	45	16	41	40
				Segment M	DQ363407.1		15	40	39
				Segment L	MK896483.1		14	39	39
Rift Valley fever virus	Phlebovirus, Bunyaviridae	(-)ssRNA	3 Segmented RNAs	Segment S	NC_014395.1	453	297	49	48
							77	45	45
				Segment M	NC_014396.1				
							79	44	43
				Segment L					
					NC_014397.1				
Ross River virus	Alphavirus, Togaviridae	(+)ssRNA	Single linear RNA		NC_001544.1	23		51	51
Sagiyama virus	Alphavirus, Togaviridae	(+)ssRNA	Single linear RNA		AB032553.1	2		52	52
Sandfly fever Sicilian virus	Phlebovirus, Bunyaviridae	(-)ssRNA	3 Segmented RNAs	Segment S	NC_015413.1	16	10	47	46
				Segment M	NC_015411.1		3	44	43
				Segment L	NC_015412.1		3	43	43
Sandfly fever Toscana virus	Phlebovirus, Bunyaviridae	(-)ssRNA	3 Segmented RNAs	Segment S	NC_006318.1	95	50	47	45
				Segment M	NC_006321		28	45	44
				Segment L	NC_006319.1		17	44	44
Semliki Forest virus	Alphavirus, Togaviridae	(+)ssRNA	Single linear RNA		NC_003215.1	10		53	52
Sindbis virus	Alphavirus, Togaviridae	(+)ssRNA	Single linear RNA		NC_001547.1	194		51	50
St. Louis encephalitis virus	Flavivirus, Flaviviridae	(+)ssRNA	Single linear RNA		NC_007580	14		50	49
Tick-borne encephalitis virus	Flavivirus, Flaviviridae	(+)ssRNA	Single linear RNA		NC_001672.1	190		54	53
Tick-borne powassan virus	Flavivirus, Flaviviridae	(+)ssRNA	Single linear RNA		NC_003687	2		53	53
Usutu virus	Flavivirus, Flaviviridae	(+)ssRNA	Single linear RNA		NC_006551.1	159		51	50
Uukuniemi virus	Phlebovirus, Bunyaviridae	(-)ssRNA	3 Segmented RNAs	Segment S	NC_005221.1	24	8	50	49
				Segment M	NC_005221		10	48	47
				Segment L	NC_005214.1		6	47	46
Venezuelan equine encephalitis virus	Alphavirus, Togaviridae	(+)ssRNA	Single linear RNA		NC_001449.1	127		50	49
Vesicular stomatitis virus strain Indiana	Vesiculovirus, Rhabdoviridae	(-)ssRNA	Single linear RNA		NC_001561	39		42	41
Vesicular stomatitis virus non-Indiana strains			Single linear RNA		MT094111.1	72		40	39
West Nile virus	Flavivirus, Flaviviridae	(+)ssRNA	Single linear RNA		NC_009942.1/1	1840		51	48
Western equine encephalitis virus	Alphavirus, Togaviridae	(+)ssRNA	Single linear RNA		NC_003908.1	38		49	49
Yellow fever virus	Flavivirus, Flaviviridae	(+)ssRNA	Single linear RNA		NC_002031	246		50	50
Zika virus	Flavivirus, Flaviviridae	(+)ssRNA	Single linear RNA		NC_012532	556		51	48

**Table 2 ijms-24-09523-t002:** Arboviruses PQSs frequency. The table reports the analysed viruses in alphabetical order. Columns indicate the virus name (Virus), the viral Genus/Family each virus belongs to (Genus, Family), the viral genomic nucleic acid type (Genome), the viral genomic structure (Genome structure), the name of each segment in case of segmented genomes (Segments), the total number of PQSs, the number of canonical (no bulges) and the number of bulged PQSs predicted in viral genomes (PQSs in viral genomes, Canonical PQSs and Bulged PQSs, respectively), the percentage of bulged PQSs on the total number of predicted PQSs (% bulged PQSs), The percentage of bulged and canonical PQSs conserved in more than 80% of analysed viral genomes (% conserved bulged PQSs and % conserved canonical PQSs, respectively), the rounded average total number of predicted PQSs on shuffled genomes and the statistical significance of the difference between the number of viral vs. shuffled PQSs (PQSs in shuffled genomes and *p*-values PQSs viral vs shuffled genomes, respectively). The symbols (**↑**), (**↓**) and (**=**) indicate that the number of PQSs predicted in the viral genomes is higher, lower or equal to the number of PQSs predicted in the shuffled genomes. Colors correspond to dsRNA (grey), negative-strand RNA ((-)ssRNA, yellow), positive-strand RNA ((+)ssRNA, blue) viruses. In segmented RNA viruses, the viral segments (S, M and L) are ordered by length.

Virus	Genus, Family	Genome	Genome Structure	Segments	PQSs in Viral Genomes	Canonical PQSs in Viral Genomes	Bulged PQS	% Bulged PQSs	% Conserved Bulged PQSs	% Conserved Canonical PQSs	PQSs in Shuffled Genomes	*p*-Values PQSs Viral vs. Shuffled Genomes
Australian bat lyssavirus	Lyssavirus, Rhabdoviridae	(-)ssRNA	Single linear RNA		45 (**↑**)	34	11	24	0	2.94	40	1.150 × 10^−30^
Banna virus	Seadornavirus, Reoviridae	dsRNA	12 Segmented RNAs	Segment 1	2 (**↓**)	2	0	0		0	6	1.15 × 10^−33^
				Segment 2	4 (**↓**)	3	1	25	0	0	6	1.66 × 10^−13^
				Segment 3	6 (**↑**)	4	2	33	0	0	5	5.23 × 10^−45^
				Segment 4	1 (**↓**)	1	0	0		0	4	3.12 × 10^−37^
				Segment 5	2 (**↓**)	0	2	100	50	0	4	2.54 × 10^−26^
				Segment 6	3 (**↓**)	3	0	0		0	5	1.90 × 10^−37^
				Segment 7	2 (**=**)	2	0	0		0	2	1.13 × 10^−4^
				Segment 8	1 (**↓**)	1	0	0		0	3	3.68 × 10^−23^
				Segment 9	0 (**↓**)	0	0	0		0	1	6.77 × 10^−24^
				Segment 10	2 (**↑**)	1	1	50	0	0	1	1.74 × 10^−3^
				Segment 11	3 (**↑**)	3	0	0		0	1	2.73 × 10^−3^
				Segment 12	2 (**↑**)	1	1	50	0	0	1	6.55 × 10^−18^
Barmah Forest virus	Alphavirus, Togaviridae	(+)ssRNA	Single linear RNA		74 (**↑**)	61	13	18	62	56	59	1.24 × 10^−43^
Bunyamwera virus	Orthobunyavirus, Bunyaviridae	(-)ssRNA	3 Segmented RNAs	Segment S	4 (**↑**)	3	1	25	0	0	2	4.16 × 10^−32^
				Segment M	5 (**=**)	3	2	40	0	0	5	0.65
				Segment L	5 (**↑**)	4	1	20	0	0	4	2.92 × 10^−4^
Bunyavirus La Crosse	Orthobunyavirus, Bunyaviridae	(-)ssRNA	3 Segmented RNAs	Segment S	4 (**↑**)	3	1	25	0	67	2	4.95 × 10^−22^
				Segment M	6 (**↓**)	3	3	50	0	33	7	3.01 × 10^−5^
				Segment L	8 (**↑**)	7	1	13	0	0	6	1.83 × 10^−8^
Bunyavirus snowshoe hare	Orthobunyavirus, Bunyaviridae	(-)ssRNA	3 Segmented RNAs	Segment S	3 (**↓**)	2	1	33	0	50	4	3.54 × 10^−4^
				Segment M	9 (**↑**)	7	2	22	50	0	7	3.72 × 10^−10^
				Segment L	5 (**↑**)	2	3	60	0	0	4	3.54 × 10^−4^
Chandipura virus	Vesiculovirus, Rhabdoviridae	(-)ssRNA	Single linear RNA		46 (**↑**)	35	11	24	18	51	30	6.10 × 10^−54^
Chikungunya virus	Alphavirus, Togaviridae	(+)ssRNA	Single linear RNA		65 (**↓**)	54	11	17	27	28	69	4.81 × 10^−9^
Crimean-Congo hemorrhagic fever virus	Nairovirus, Bunyaviridae	(-)ssRNA	3 Segmented RNAs	Segment S	6 (**=**)	4	2	33	0	0	6	0.96
				Segment M	23 (**↑**)	18	5	22	0	0	16	1.53 × 10^−41^
				Segment L	18 (**↓**)	16	2	11	0	0	26	1.70 × 10^−32^
Dengue virus 1	Flavivirus, Flaviviridae	(+)ssRNA	Single linear RNA		61 (**↑**)	52	9	15	0	15	49	7.80 × 10^−38^
Dengue virus 2		(+)ssRNA	Single linear RNA		64 (**↑**)	53	11	17	9	15	44	6.40 × 10^−55^
Dengue virus 3		(+)ssRNA	Single linear RNA		69 (**↑**)	54	15	22	7	20	49	6.05 × 10^−59^
Dengue virus 4		(+)ssRNA	Single linear RNA		77 (**↑**)	64	13	17	31	19	52	4.29 × 10^−70^
Dhori virus	Thogotovirus, Orthomyxoviridae	(-)ssRNA	6 Segmented RNAs	Segment 1	10 (**↑**)	8	2	20	0	0	8	5.81 × 10^−14^
				Segment 2	5 (**↓**)	4	1	20	0	0	7	1.18 × 10^−12^
				Segment 3	7 (**↑**)	7	0	0		0	6	3.73 × 10^−6^
				Segment 4	11 (**↑**)	10	1	9	0	0	7	6.27 × 10^−37^
				Segment 5	4 (**↓**)	4	0	0		0	7	1.10 × 10^−20^
				Segment 6	5 (**=**)	2	3	60	0	0	5	0.22
Dugbe virus	Nairovirus, Bunyaviridae	(-)ssRNA	3 Segmented RNAs	Segment S	5 (**=**)	4	1	20	0	0	5	0.45
				Segment M	12 (**=**)	12	0	0		25	12	0.09
				Segment L	18 (**↑**)	15	3	17	0	40	5	0.45
Eastern equine encephalitis virus	Alphavirus, Togaviridae	(+)ssRNA	Single linear RNA		59 (**↓**)	47	12	20	58	68	61	1.93 × 10^−3^
Isfahan virus	Vesiculovirus, Rhabdoviridae	(-)ssRNA	Single linear RNA		33 (**↑**)	22	11	33	100	100	27	4.34 × 10^−22^
Japanese encephalitis virus	Flavivirus, Flaviviridae	(+)ssRNA	Single linear RNA		101 (**↑**)	82	19	19	0	11	73	8.61 × 10^−53^
Langat virus	Flavivirus, Flaviviridae	(+)ssRNA	Single linear RNA		125 (**↑**)	98	27	22	33	43	113	2.22 × 10^−34^
Louping ill virus	Flavivirus, Flaviviridae	(+)ssRNA	Single linear RNA		130 (**↑**)	106	24	18	33	37	114	4.51 × 10^−42^
Mayaro virus	Alphavirus, Togaviridae	(+)ssRNA	Single linear RNA		66 (**↓**)	52	14	21	7.	0	70	2.49 × 10^−12^
Murray Valley encephalitis virus	Flavivirus, Flaviviridae	(+)ssRNA	Single linear RNA		87 (**↑**)	79	8	9	0	14	66	4.61 × 10^−55^
O’nyong-nyong virus	Alphavirus, Togaviridae	(+)ssRNA	Single linear RNA		53 (**↓**)	42	11	21	36	45	59	4.84 × 10^−17^
Oropouche virus	Orthobunyavirus	(-)ssRNA	3 Segmented RNAs	Segment S	4 (**↑**)	4	0	0	0	16	3	4.99 × 10^−10^
				Segment M	2 (**↓**)	2	0	0		50	4	1.40 × 10^−15^
				Segment L	2 (**↓**)	2	0	0	0	0	5	2.23 × 10^−29^
Punta Toro phlebovirus	Phlebovirus, Bunyaviridae	(-)ssRNA	3 Segmented RNAs	Segment S	4 (**=**)	2	2	50	0	6	4	7.81 × 10^−2^
				Segment M	6 (**↓**)	3	3	50	0	0	8	3.06 × 10^−10^
				Segment L	12 (**↓**)	11	1	8	60	50	13	0.11
Rift Valley fever virus	Phlebovirus, Bunyaviridae	(-)ssRNA	3 Segmented RNAs	Segment S	8 (**↓**)	7	1	13	86	50	9	2.84 × 10^−7^
					16 (**=**)	11	5	31	0	17	16	0.64
				Segment M								
				Segment L	24 (**↑**)	17	7	29	0	0	21	4.57 × 10^−12^
Ross River virus	Alphavirus, Togaviridae	(+)ssRNA	Single linear RNA		76 (**↓**)	61	15	20	33	46	77	1.61 × 10^−2^
Sagiyama virus	Alphavirus, Togaviridae	(+)ssRNA	Single linear RNA		71 (**↓**)	55	16	23	100	98	86	2.46 × 10^−43^
Sandfly fever Sicilian virus	Phlebovirus, Bunyaviridae	(-)ssRNA	3 Segmented RNAs	Segment S	7 (**↓**)	6	1	14	0	0	9	2.03 × 10^−8^
				Segment M	17 (**↑**)	12	5	29	60	50	14	6.75 × 10^−13^
				Segment L	23 (**↑**)	16	7	30	86	50	20	5.13 × 10^−13^
Sandfly fever Toscana virus	Phlebovirus, Bunyaviridae	(-)ssRNA	3 Segmented RNAs	Segment S	7 (**↓**)	6	1	14	0	17	8	1.04 × 10^−6^
				Segment M	17 (**↑**)	12	5	29	0	0	15	7.39 × 10^−10^
				Segment L	21 (**↓**)	16	5	24	0	6.25	22	0.13
Semliki Forest virus	Alphavirus, Togaviridae	(+)ssRNA	Single linear RNA		95 (**↑**)	79	16	17	88	87	92	6.37 × 10^−4^
Sindbis virus	Alphavirus, Togaviridae	(+)ssRNA	Single linear RNA		68 (**↓**)	52	16	24	75	69	76	1.89 × 10^−21^
St. Louis encephalitis virus	Flavivirus, Flaviviridae	(+)ssRNA	Single linear RNA		85 (**↑**)	72	13	15	23	10	70	1.27 × 10^−37^
Tick-borne encephalitis virus	Flavivirus, Flaviviridae	(+)ssRNA	Single linear RNA		120 (**↑**)	99	21	18	0	1	111	2.26 × 10^−23^
Tick-borne powassan virus	Flavivirus, Flaviviridae	(+)ssRNA	Single linear RNA		123 (**↑**)	101	22	18	100	100	102	4.25 × 10^−49^
Usutu virus	Flavivirus, Flaviviridae	(+)ssRNA	Single linear RNA		92 (**↑**)	72	20	22	50	72	79	7.96 × 10^−42^
Uukuniemi virus	Phlebovirus, Bunyaviridae	(-)ssRNA	3 Segmented RNAs	Segment S	8 (**↓**)	6	2	25	0	0	10	4.64 × 10^−14^
				Segment M	16 (**=**)	10	6	38	0	10	16	0.22
				Segment L	30 (**↑**)	29	1	3	100	66	28	1.85 × 10^−6^
Venezuelan equine encephalitis virus	Alphavirus, Togaviridae	(+)ssRNA	Single linear RNA		63 (**↓**)	49	14	22	0	2	69	1.17 × 10^−18^
Vesicular stomatitis virus strain Indiana	Vesiculovirus, Rhabdoviridae	(-)ssRNA	Single linear RNA		34 (**↑**)	25	9	26	33	48	26	1.98 × 10^−35^
Vesicular stomatitis virus non-Indiana strains			Single linear RNA		29 (**↑**)	22	7	24	86	95	20	5.57 × 10^−40^
West Nile virus	Flavivirus, Flaviviridae	(+)ssRNA	Single linear RNA		88 (**↑**)	75	13	15	38	40	81	5.40 × 10^−17^
Western equine encephalitis virus	Alphavirus, Togaviridae	(+)ssRNA	Single linear RNA		55 (**↓**)	42	13	24	69	71	64	4.96 × 10^−25^
Yellow fever virus	Flavivirus, Flaviviridae	(+)ssRNA	Single linear RNA		94 (**↑**)	78	16	17	0	5.	73	1.77 × 10^−52^
Zika virus	Flavivirus, Flaviviridae	(+)ssRNA	Single linear RNA		101 (**↑**)	84	17	17	18	12	79	2.12 × 10^−56^

## Data Availability

Data are available in Appendix A.

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
