# Peer review of "Presence, Location and Conservation of Putative G-Quadruplex Forming Sequences in Arboviruses Infecting Humans"

_ijms, 2023, doi:10.3390/ijms24119523_

Round 1

Reviewer 1 Report

In the past few years, exploiting guanine quadruplexes (G4s) as potential therapeutic targets has become increasingly interesting. In this work, the authors focused on putative G4 forming sequences (PQSs) in human arboviruses predicted by "pqsfinder" and reported the presence, conservation, and localization of PQSs. Based on the PQS prediction from more than twelve thousand viral genomes, they found that the abundance of PQSs in arboviruses depends on the type of nucleic acid that constitutes the viral genome. Specifically, positive-strand ssRNA arboviruses, especially Flaviviruses, are significantly enriched in highly conserved PQSs, which are located in coding regions and at genome ends. The authors predicted bulged PQSs by modifying parameters in "pqsfinder", accounting for 10-17% of the total predicted PQSs. This paper is well-written, providing insights into unexplored aspects of arbovirus biology and shedding light on innovative anti-arbovirus targets.

 However, there are a few areas that could benefit from improvement. Firstly, the authors need to explain how they calculated PQS density. Additionally, is there any correlation between PQS density, its score, and conservation? Secondly, the authors found that many positive-strand ssRNA viruses have highly conserved PQSs throughout the genome. It would be very interesting to test if there is any difference regarding conservation between the predicted canonic and bulged PQSs. Thirdly, the resolutions for Figures 1 and 2 are too low, making it hard to read the text in the figures.

 Other minor comments:

 Webpages for reference #1 and #8 cannot be found.

There are several typos in Table 2: 1) "% bulged PQSs" in Banna virus (Segment 1) should be 50%; 2) The numbers in Rift Valley fever virus need to be checked.

The statement in lines 268-269 is too strong without experimental data.

Reviewer 2 Report

In this study the authors extend their former study on PQSs from human viruses on the arboviruses. They identify PQSs traits specific for CDSs and 3'-UTRs. The data seem strong with a decorrelation between sequence conservation in different virus strains and PQSs  loci. Figures 1 and 2 are very hard to read. A subset of the graphs may be selected to illustrate better and send the remaining parts to the supplementary data section.
